# Feasibility Study of the Secondary Level Active School Flag Programme: Study Protocol

**DOI:** 10.3390/jfmk4010016

**Published:** 2019-03-26

**Authors:** Kwok W Ng, Fiona McHale, Karen Cotter, Donal O’Shea, Catherine Woods

**Affiliations:** 1Department of Physical Education and Sport Sciences; Centre of Physical Activity and Health Research; Health Research Institute, University of Limerick, V94 T9PX Limerick, Ireland; 2School of Educational Sciences and Psychology, University of Eastern Finland, 80101 Joensuu, Finland; 3Active Schools Flag, Mayo Education Centre, F23 HX48 Castlebar, Ireland; 4St. Vincent’s University Hospital, University College Dublin, D04 T6F4 Dublin, Ireland

**Keywords:** physical activity, adolescent, health promotion, activePal, intervention

## Abstract

Taking part in regular physical activity (PA) is important for young adolescents to maintain physical, social and mental health. Schools are vibrant settings for health promotion and the complexity of driving a whole-school approach to PA has not been tested in the Irish school context. The feasibility of the pilot programme of the Department of Education and Skills second level Active School Flag (SLASF) is needed. SLASF is a two year process that consists of the Active School Flag (ASF) certificate programme (year 1) and the ASF flag programme (year 2). This protocol paper is specific to the first year certificate process. Three schools around Ireland were recruited as pilot schools to carry out the year-long SLASF programme with 17 planned actions involving the entire school. Students in the transition year programme have a particular role in the promotion of PA in SLASF. Data collection consists of physical measures, accelerometers, survey data and interviews at the beginning and the end of the academic year. The primary focus on the feasibility of the programme is through process evaluation tools and fidelity checks consisting of implementation of the SLASF programme through whole-school surveys, focus group discussions of key stakeholder groups, as well as one-to-one interviews with a member of management at each school and the SLASF coordinator of the school. Secondary outcomes include PA levels and its social cognitive theories based correlates through physical health measures, surveys carried out pre- and post-intervention, as well as focus group discussions of the students. The results of this study are needed to improve the development of the SLASF through a predetermined stopping criteria and inclusion into systems thinking approaches such as the Healthy Ireland Demonstration Project.

## 1. Introduction

There are multiple reasons—physical, psychological, social, environmental—health for adolescents to take part in regular physical activity (PA). However, adolescence forms a highly volatile stage in life where transitional periods can influence behaviour [1] and is a critical time for PA participation where habits—good or bad—developed, later persist into adulthood [2]. Despite the evidence of health benefits from PA, a clear reduction in PA levels is apparent alongside increasing age in adolescence. Data from the Children’s Sport Participation and Physical Activity (CSPPA) study highlighted a decline in meeting the PA guidelines, at least 60 min of moderate-to-vigorous (MVPA) per day, from 18% among 12y olds (first year of second level education in Ireland) to 6% among 18y olds (last year of second level education in Ireland) [3]. Similarly, a systematic review and pooled analysis indicated that PA levels decline by a mean of 7% per year over this period [4,5]. By the age of 15y, on average, across 42 countries in the Health Behaviour in School-aged Children study, only 11% of girls and 21% of boys self-reported sufficient MVPA levels [6]. Global PA decline throughout adolescence, across a range of measurement units, appears as high as 60–70% [4]. Action is needed to reduce the drop in PA levels.

Actions that target the drop in PA levels can include interventions at the individual, community and within policy [7]. Although policy interventions may provide the best return on investment [8], creating such changes requires a strong evidence base that interventions targeting the behaviours actually works. At the individual level, interventions that were designed to increase school-based PA levels have yet to demonstrate its effectiveness [8]. In particular, multi-component PA interventions have yet to demonstrate improvements in overall MVPA [9]. However, at the community level, school-based PA programmes have demonstrated some positive effect on overall PA levels [10]. 

### 1.1. Whole-School Approaches to Physical Activity Promotion

Successfully run school-based PA programmes includes quality physical education, physical activities before-, during- and after-school, at the school grounds, as well as activities based in the local community [11]. Haapala and colleagues [11] suggested that recess time activities before, during and after school are opportune moments to promote PA. Similarly, a study based on a 12 week walking intervention among girls who took part before-school and during recess times increased the time in light intensity PA by 10 min but differences in MVPA were not statistically significant when compared to the intervention group [12]. Some recent studies have investigated the role of peers in boosting PA levels. For example, peer leaders (15–16y olds) slightly older than the target group (13–15y olds) encouraged after-school activities. By the end of the 7-week after-school intervention, there was a statistically significant increase of three minutes of daily MVPA [13]. Same-aged peers were also effective in promoting six minutes of daily MVPA through diffusion messages among girls during the entire school day [14]. Statistically significant changes on PA levels were found for those identified as low active at baseline in an intervention that focused on increasing step count in both boys and girls [15]. Few school-based programmes managed to succeed in increasing boys’ PA levels for school-based interventions through a peer-led model [16,17]. Programmes with in-class activities that promote PA by the class teachers are another way to increase PA levels [18]. The latter activities requires a whole-school approach towards PA promotion and in Ireland, this is known as the Active School Flag (ASF) [19].

The ASF (www.activeschoolflag.ie) is a Department of Education and Skills (DES) initiative supported by Healthy Ireland. It takes a ‘whole-school approach’ and requires all members of the school community to work together to strengthen the delivery of the physical education programme and to promote PA throughout the school in a fun and inclusive manner. It emphasises quality PE, co-curricular PA as well as partnerships with students, parents and the wider community. In 2018, 29% (*N* = 1329) of primary schools in Ireland had achieved ASF status. Some have the need for renewal, thus at the time of print, 722 primary schools currently have possession of an Active Flag. The journey towards achieving this status begins with self-evaluation in the areas of (i) physical education (ii) PA and (iii) partnerships. Following this, schools are expected to devise a strategic plan and implement changes to help improve the quality of PE, provide additional opportunities for PA and support additional involvement with the wider community. Lastly, the school has to provide a week-long focus on PA in an Active School Week.

The ASF is well-established as a primary school initiative but has yet to be developed for the secondary level education sector. A generic model has been tried among secondary level schools, although it did not achieve the same effect because the uptake of secondary level schools is below 5%. The most notable differences were the lack of whole-school engagement in the secondary level schools. Feedback from those experiences suggested that the generic ASF in secondary level was viewed as a physical education programme as opposed to a whole-school initiative. Furthermore, there are structural differences between primary and secondary level schools, thus a direct translation of the primary ASF to the secondary level setting would be specious. Some examples of differences include; in primary schools, a single teacher generally remains with the same class throughout the entire day, whereas at secondary level schools, students are taught by multiple teachers. In the secondary level schools, many schools do not have a compulsory timetabled physical education in accordance with DES recommendations (i.e., a double timetabled period for all students every week). Secondary level schools tend to be larger and cater for more students than primary schools, yet facilities for promoting PA can be limiting. Peer influences tend to be stronger at secondary level education [20] and students will often choose to socialise with their friends and be sedentary rather than take part in PA [21]. In primary schools, all teachers are encouraged to take part in continuous professional development (CPD) as part of their classroom responsibility in the areas of physical education and PA. Class teachers in secondary level schools tend to be specialists in their subject area and may not have the confidence or competence to create active breaks in the classroom [22]. 

In the majority of Irish secondary level schools, there is the ‘Transition Year’ (TY) period, whereby it is non- examinable and students may go on to try out activities beyond the normal school curriculum [23]. The aims of TY are to “increase social awareness and social competence with ‘education through experience of adult and working life’ as a basis for personal development and maturity” [24]. TY programmes can cover various aspects of personal development, including visits to hospitals to learn about health behaviours [25], as well as taking part in other science programmes [26]. Many TY based programmes seek to develop youth leadership skills (i.e., Gaisce the President’s Award, Young Social Innovators, Gaelic Athletic Association (GAA) future leaders, etc.). 

### 1.2. Theoretical Background to the Study

The theoretical premise of this feasibility study is underpinned by social cognitive theories [27] These theories are the most cited for health behaviour change interventions [28]. In these theories, there is a triad relationship between personal and environmental factors with behaviours being explored through individual level factors such as self-efficacy, stages of readiness for behavioural change and school related autonomy. In self-efficacy theory [29] there are four sources; mastery accomplishments, vicarious experiences, verbal persuasion and physiology are strong predictors of behaviour. Novice learners, such as students in TY, tend to have low confidence to promote PA at the beginning of the year [30]. When schools express their interest to be an SLASF school, a resource pack (see methods section) would support these students to improve the self-efficacy (opportunities for mastery accomplishments), create timetabled meetings with the purpose to focus on the SLASF (opportunities for vicarious experiences) and regular contact with others who are also carrying out the SLASF (opportunities for verbal persuasions), the sources of self-efficacy may create a better suited environment for promoting PA. 

Being autonomous in making decisions is known to be directly related to intrinsic motivations towards a behaviour [31]. However, in the school context, students may feel their overall autonomy is restricted to the bounds of the school. As a result, some students may find the school environment something that they can thrive in, whereas others may feel the environment restricts individual choices. The autonomy supportive environment is as important as having autonomy among school-aged children because it can lead to greater levels of PA outside of the school context [32]. Factors that may influence the autonomy or autonomy support can include the perception of school academic performance and the way teachers and peers support academic performance [33], perceptions of ability to make decisions in the school [34] and feelings of belonging in the school [35]. Furthermore, it has been reported that individual well-being is associated with individual’s sense of autonomy [36], therefore studies may need to consider the mediating factors of student well-being. 

The levels of readiness for taking part in regular PA would vary vastly among the students throughout the school. According to the transtheoretical model [37] individuals are at different stages of their intention for behavioural change. Progression between the five noted stages include pre-contemplation (not ready), contemplation (getting ready), preparation (ready), action and maintenance. According to the model, the factors that influence moving between these stages consist of biopsychosocial factors, including self-efficacy, own and socially supported decisions and going through the process of change [38]. Through awareness of the stages of change among the target population, there is a greater understanding of how to create targeted interventions for the promotion of PA.

### 1.3. Measures of Feasibility 

The SLASF programme is in its pilot phase as part of a larger systems based Healthy Ireland Demonstration Project. It is also a feasibility study, because we aim to investigate whether it is suitable in secondary level schools and if so, how to do that [39]. According to Bowen and colleagues [40], designing feasibility studies can have eight areas of focus; acceptability, demand, implementation, practicality, adaptation, integration, expansion and limited efficacy. Although the SLASF is a non-clinical study, the feasibility study is useful to provide an indicator for stopping, revisions or continuing for a scaling up a randomised control trial [41]. Moreover, the predefined rules need to be put in place prior to the analysis to prevent uncontrollable biases [42]. In Table 1, Bowen and colleagues’ areas of foci for this feasibility study are broken down into the areas which this study researches and stopping criteria.

### 1.4. Purpose of the Study

The primary objective for this study is to see if the SLASF certificate is an acceptable programme in secondary level Irish schools. The secondary objectives are to see how feasible it is to operationalise the components needed for testing a year-long intervention. These include collecting accelerometer and physical health measures of students in the schools, completion of survey questions for pre- and post-intervention evaluation and investigate areas for improvement from a pilot study to scale-up intervention programme. 

## 2. Methods and Design

### 2.1. Setting

Eligible schools include secondary level schools in the Republic of Ireland that have not previously applied for the SLASF. Special educational needs schools or schools without a range of students from each year group are excluded. The feasibility study will be conducted in three secondary level schools, covering the demographics of a girls only school, a mixed school with designated ‘Delivering Equality of Opportunity in Schools’ (DEIS) status and a mixed mainstream school. A DEIS school is part of the Department of Education and Skill’s action plan for educational inclusion to address disadvantaged education.

### 2.2. Recruitment

#### 2.2.1. School Recruitment

The SLASF programme is a comprehensive programme and employs a whole school approach. As an intervention programme, schools were invited to take part if they are not currently in the process of obtaining the Active School Flag. The normal process of the SLASF recruitment is for schools to make an application but this was suspended to allow time to develop the new model. Schools that were interested in carrying out the feasibility study contacted the Active Schools office and requested to pilot the second level programme. There were three schools that responded and accepted because they each have a strong well-being structure in place in their schools. This is a feasibility study with secondary outcomes on efficacy hence matched control schools (one girls only, one mixed and one mixed DEIS school) were recruited to take part in the survey component of the study. No reserve list was formed for this cohort but engagement of all three schools throughout the year to follow the feasibility and evaluate the SLASF process.

#### 2.2.2. Student Recruitment

In the participating schools, there are two levels of student engagement—basic and comprehensive. Although all students are exposed to the same SLASF programme, those engaged in the SLASF Basic are required to complete less data collection measures than those in the SLASF Comprehensive. The whole school is involved in the SLASF Basic and a random class from each year group is selected to take part in the SLASF Comprehensive. Classes are known as mixed ability tutor groups. Students obtain their consent to take part in the study. Even though there may be some students who do not obtain consent or would not like to withdraw from the comprehensive aspect of the study, there is no likely reason for them to influence the feasibility of the study. 

The SLASF initiative is structured to fit into the TY framework providing real and meaningful opportunities for student voice and youth leadership. A TY class would be nominated by the school as the SLASF TY class. This TY class attends classes in relation to SLASF tasks including youth leadership, mentoring and PA promotion. 

### 2.3. Consent

All students are given information letters for their parents. The information stated the purpose of the study as well as components of the study that would involve their child. Also, the letter invites the parents to speak to their children about the study and there are contact details to the research team to answer questions the parents may have. Because of population bias from active consent, passive consent was used for all students in the basic part of the programme. The school manages the returned forms of withdrawal. This is because it is part of the whole-school process and as a feasibility of programme, schools need to be aware of how to handle withdrawal from the programme. The online survey requests the students to give their assent (if under the age of 16) or consent (if over the age of 16). 

Students assigned to the comprehensive study, receive the same information sheets as the basic programme but in addition, information about the other measures that are included as part of the feasibility and process evaluation. Parents are asked to give opt in consent due to the nature of data collection (i.e., use of accelerometers). As part of Irish law, students over the age of 16 can give their own consent to take part in the study. All students were asked to opt in to be included in video recordings of interviews.

### 2.4. Allocation Strategy

There was no allocation strategy used to select the schools. As a feasibility study, it was important to test the SLASF in various contexts. Three contexts were chosen; a DEIS school, a girls only school and a mixed school. All schools are considered to be large in size and this would test out the possible whole-school approach. 

### 2.5. Active School Flag Intervention

The SLASF feasibility model is co-designed by a SLASF steering group and staff, researchers of the University of Limerick and feedback from the three lead schools. The SLASF process is designed to be peer-led by a TY SLASF class, who will have the support of an SLASF coordinator, SLASF committee members, school staff and school management. The initiative challenges peers to find ways to encourage more students in their school to engage in school-based PA opportunities (year 1—SLASF certificate) and community-based PA opportunities (year 2—SLASF flag). During year 1 (Active School Flag certificate) the focus is on increasing participation in school-based PA opportunities. Year 2 (Active School Flag) is focused on community-based activities. 

Previously the SLASF was viewed as a physical education initiative. In order to generate greater whole-school engagement, the SLASF tasks are formatted to draw support for the SLASF TY team from school management and teachers across a variety of different subjects. The new format of the SLASF process complements two current key educational initiatives: The Well-Being Framework by the Irish National Council for Curriculum and Assessment, the School Self-Evaluation process by the Department of Education and Skills and a new initiative presently at draft stage: The Parent and Student Charter also by the Department of Education and Skills. Students working towards Gaisce the President’s Award can use their SLASF work to fulfil their Community or Personal Skill challenge requirement. Another benefit of SLASF is that it will link schools with the current national Healthy Ireland PA programmes and national youth charity events including: Get Ireland Walking—www.getirelandwalking.ieParkrun—www.parkrun.ieGet Ireland Swimming—www.swimireland.ie/get-swimmingSwim for a Mile Challenge—www.swimireland.ie/get-swimming/swim-for-a-mileDarkness into Light—www.darknessintolight.ieCycle Against Suicide—www.cycleagainstsuicide.com

The SLASF programme is a whole-school approach to increase PA opportunities and generate opportunities for student voice and youth leadership. Currently, there are two levels. The first level is a certificate. This is open to all secondary level schools. It can serve as a good link from primary schools to continue on the ethos of active schools and allow a school to consider whether to take the next step or not. Moreover, SLASF is a DES (Department of Education and Skills) initiative and it can only be awarded to schools that adhere to physical education timetable recommendations that is, a double timetabled period of physical education for all year groups. This provision is not in place in a large number of secondary level schools, thus heretofore excluding them from the SLASF process. The introduction of the certificate level, without eligibility criteria, opens the SLASF process to all interested post primary schools. If the SLASF certificate proves beneficial it may encourage them to revisit their timetable policy. 

Achievement of the flag is a whole school process, meaning that management, staff and students all play a role in the programme. There is a requirement for website updates and an online presence. In order to achieve the SLASF Certificate a number of tasks must be completed during year 1. These include: (1) a staff slideshow, (2) an SLASF team slideshow, (3) class time slideshow (4) SLASF training day, (5) an SLASF awareness week (towards the beginning of the year), (6) website showcase, (7) SLASF whole school questionnaire, (8) SLASF launch event, (9) SLASF action plan, (10) ‘Did You Know?’ campaign, (11) PA module as part of Social, personal and health education (SPHE) subject for junior cycle students, (12) Active School WALKWAY, (13) Community Mapping of extra curriculum activities, (14) Community Event, (15) Active School Week, (16) SLASF accreditation visit and (17) school PA space audits (Figure 1).

All 17 activities are scheduled throughout the school year including a combination of staff and students as the main actors in this process. The SLASF TY class should take leadership guided and supported by the SLASF coordinator and committee on the programme. A key part of the process is that the SLASF coordinator has timetable provision allocated to two class periods per week to carry out work with the TY class. The committee includes staff representatives from the school, including one from management, an SLASF coordinator, two other staff who work on the well-being curriculum to include SPHE, Civic, Social, Political Education (CSPE) and Physical Education teachers, as well as four youth leaders. Structure of different actors in the process can be viewed in Figure 2.

Schools wishing to work towards the second level of the SLASF process, the Active School Flag, must have completed the certificate level and be able to confirm that they timetable physical education in accordance with DES recommendations.

At the student level, there are three levels of involvement. There is the SLASF team, which is comprised of the SLASF TY programme group. There are four SLASF youth leaders who represent the team at the school committee level. To be a youth leader, the SLASF team member needs to apply for the position through an application process that is evaluated by the staff members of the SLASF committee. The youth leaders end up representing the student voice at the SLASF committee and have the responsibility of presenting the SLASF action plan and the SLASF end of year review to the school Principal. The third student level is the SLASF class representatives. There are two in every class in the school and students can also apply for this position through an application form. The selection will be made by the class tutor in consultation with the SLASF coordinator. 

#### 2.5.1. TY Leader Role

The TY leader role is responsible for planning, promoting and implementing the SLASF initiatives and events throughout the school year. Based on self-efficacy theory [29], TY leaders are a closer connection to the students in the schools than teachers, thus strengthen vicarious experiences. Moreover, social support and leadership from the TYs can reinforce the basic premise of proactive behaviour change [43]. The SLASF team can be identified by being given pins to wear on their uniform. Moreover, part of the time spent on SLASF activities can be used as part of a time bank for other volunteering programmes, such as, Gaisce the President’s Award. The selected SLASF youth leaders will receive their own distinct pins to wear on the uniform. 

#### 2.5.2. Activities for Certification

For schools to be successful in achieving the activities needed for certification, there is a yearly planner with guideline dates for task completion that the schools will use to keep on track. This also includes the accreditation visit. For example, the SLASF slideshows for the staff, team and youth leaders need to have been completed before the 2nd week of the school year. A designated training day takes place a week later. The purpose of this training day is to introduce the pilot schools to each other and the research team. The research team describes the whole year process evaluation and measurements taken throughout the year. There are co-design opportunities between the TY leaders and researchers to formulate the surveys used to collect data. The training day is not expected to run every year once feasibility is over. At this training day, two members of staff, the SLASF coordinator and another on the SLASF team take four student leaders to a training venue to learn about how to run the activities throughout the year. The website should also go live at the outset of the process. The website should encompass an easy to find link to the SLASF section of the site and there are the four core parts of the SLASF process; Physical Education; PA; Partnerships; and Active School Week.
Another activity that the school needs to complete is the SLASF Awareness Week. This should be completed two weeks after the training day. School census questionnaire is deployed a week after the awareness week and this precedes the official launch of the SLASF process.During the launch, there would be a school wide tug of war (TOW) competition that is planned, promoted and organised by the SLASF committee. For the launch day the overall winning TOW team will compete against a staff team at a whole school event to launch the SLASF initiative.

A school census questionnaire was developed to help the SLASF team to identify their action plan. Core questions about PA opportunities, physical education, involvement in extra-curriculum activities and barriers to PA were included in an online survey. For a week after the awareness week, the survey is available for completion. All responses are anonymous and completed confidentially. Class teachers supervise and help answer any technical questions related to the completion of the survey. The data is stored on a secure server that is only accessible to the researchers. However, the overall results for each variable would be computed and provided for the school to carry out their own descriptive analyses with the TY group. The TY class can then produce meaningful findings from the survey to show the school through the notice board and used for one of the planned actions. 

For the TOW event a free rope and a 2 ½ h workshop will be provided for each school. This will enable TYs to coach and officiate TOW competitions and SLASF committee members that complete the course will receive a TOW Community Coach certificate upon completion. Each class in the school will be involved in this event with each having 3 TOW teams that compete against each other during tutor time or physical education class to decide what team will represent the class. Each class TOW team competes against the other classes in their year group during lunchtime to find the best TOW team for their year. Local role models can be invited to help launch the event by taking part in one of the TOW teams. This event should take place before the mid-autumn break. 

After the mid-autumn break, the schools would have access to their school’s results from the school questionnaire. The SLASF teams are given a month to review the results and start to design an SLASF action plan. At least three action points need to be agreed upon by the SLASF team. The proposed SLASF actions should be presented to the Principal for agreement. The agreed actions are then implemented in the second half of the school year. Towards the end of the year, the three agreed actions will be reviewed by the SLASF team members and presented to the school Principal during the last two weeks of the school year.

In addition, all students in the selected year group in the school will take part in a four weeks PA module delivered by the Social and Personal Health Education (SPHE) teachers. There would have to be a ‘Did You Know?’ campaign around the school that helps raise awareness about the benefits of PA for teenagers, in particular the positive impact that PA has upon focus, concentration and academic achievement. Another practical task for the SLASF team is to signpost an Active School WALKWAY. The walkway is a route that can be used by the students in the school during recess time or under teacher supervision for active learning activities, before/after tests or during free classes. SLASF, in partnership with Get Ireland Walking, designed Active School WALKWAY packs consisting of colourful outdoor all-weather sign post plaques which include orienteering symbols. One of the tasks that the SLASF TY class have to undertake is to map, measure and erect the walkway signposts to create a school walking route. A school WALKWAY Day where all classes get the opportunity to complete the walkway route with their teachers on a nominated school day needs to be agreed by school management. Then it is organised and promoted by the SLASF TY team. The organised walkway can be used as part of orienteering activities during timetabled physical education, as well as other school-based initiatives. 

As the year ends, the school prepares for the accreditation visit for the certificate. A follow up visit takes place during the acquisition of the flag year. Prior to this, the school needs to organise a community mapping exercise and community events which should help with the design of the Active School Week (ASW) programme. The main aims of the ASW are to promote PA in a fun and inclusive way, as well as raising awareness about the availability and variety of PA opportunities for teenagers and their families in their local community. Throughout this week the school provides many and varied opportunities for staff and students to become more physically active throughout the school day.

#### 2.5.3. Expected Outcomes Tables and Measures

According to their training resources, the programme aims to impact on a number of areas. However, to measure them all, multiple sources are required. A collection of survey instruments can be used to measure some of the outcomes, whereas some interviews can be used to evaluate other outcomes. In addition, the programme is year-long and a whole-school approach, hence site visits and checking on progress through logbook entries would be used to determine the processes carried out during the study. In Table 2, there is a list of the areas that SLASF aims to promote and some measures that can be used to test these outcomes.

#### 2.5.4. Questionnaires

There are two types of online surveys carried out throughout the year. The basic survey is completed by the entire school. This survey is anonymous and the focus is on PA participation and barriers to school related physical activities. This survey is a compulsory part of the SLASF process. Administration of the survey is decided by the school, with the intention to cover the entire school. Ideally, a census sweep of the school takes place at the same time. However, there may be some technical issues that may prevent this from happening. For example, schools may have a limited number of computers accessing the internet at any one time (bandwidth limits), may have a limited number of units to complete the survey (lack of tablets or computers) or could not get all the school to take part at the same time (timetabling issues). The results of the survey will be given back to the school for the purpose to plan specific school-based interventions. Therefore, it is important that the mode of data collection, analysis and reporting can be completed quickly and easily. Failing all technical capabilities to collect from an online platform, extra resources would be dedicated to ensure double coding from pen and paper surveys. 

The second type of survey is a comprehensive survey, used for evaluating the feasibility of the study. The participants in this study input their user-ID so that the data can be linked from the beginning and end of the year long programme. Completion of the online survey takes place as one of the testing stations during the data collection visits. All the students have tablets or allocated to a school computer to complete the online survey. Details of the instruments are reported in Table 3.

### 2.6. Process Evaluation

#### 2.6.1. Logbook Activities

Each school is given a logbook to record their activities. This is used as part of the accreditation process and is used by the researchers to evaluate the processes that the school used. The logbook is mainly used by the SLASF team and the SLASF committees. Every week, the TY students have the opportunity to complete a small section in the diary to record what took place. The diary is linked to the school year and the expected timescale for carrying out specific activities. There is also a chart for the SLASF team to complete by recording the agreed actions to be carried out by the SLASF team. The team need to record the date of the agreed action, a short title for the action, the person(s) responsible, date of the action completed and a check box. 

The TY team carries out a brief version of the System of Observing Play and Leisure Activity in Youth (SOPLAY) [52]. SOPLAY is a direct observation tool that is used by the TYs to assess PA levels within specific PA areas in the school. Due to resources, the full SOPLAY protocol had to be reduced down to three specific areas around the school. Furthermore, the TYs use tablets to video record the specific area and retrospectively carry out the observations. It is designed in this way because the technology is more readily available in schools than the time when SOPLAY was created by McKenzie and colleagues [52]. Through, observing the video recordings, the results can be verified so the validity of the results are stronger. Trained researchers with the SOPLAY counting system can verify the results from the TYs by matching the observation results. The videos can also be used as part of a TY class, where the students can get an understanding of ways to record the different intensities of PA. The SOPLAY exercise is carried out over six times throughout the school year. The assignment of the dates are researcher assigned days. The TYs are informed of the audit in the morning of the day the recording takes place. To reduce potential bias in the results, the TYs are reminded not to tell others that they are carrying out the observation. Observations take place twice during lunchtimes, one 10 min into the beginning and the second when there is 10 minu left. 

Another activity recorded in the logbook is the SLASF committee meetings. The logbook provides space for six meetings throughout the year. The meeting minutes include the people in attendance, the areas of discussion and the actions that were agreed. There is also space in the logbook has space for a list of agreed action created by the TY class during their timetabled class time. To encourage compliance, there is room for information such as the agreed action, the person responsible and the date for completion. In addition, there is room in the logbook for the TY team and coordinator to note activities that take place in a specific week. For each week, tasks that are suggested, such as the slide show, presentation of the action plan and so forth are available for the TY and coordinator to help remind to be on track. 

#### 2.6.2. Whole-School Surveys

There are three surveys to be carried out by all students in the school. The whole-school survey is part of the feasibility study and is carried out through an online survey platform. It is a mandatory action to be carried out by the school and is carried out during the first two months of the academic year. The school uses this information for creating and implementing three school specific action plans. Within this survey there are details of participation levels and barriers to taking part in physical education and extra curriculum activities. Both staff and students complete a second survey halfway through the process with items also related to process evaluation. Items will test implementation, fidelity and satisfaction of the tasks completed to date. The final whole-school survey has items related to process evaluation and is completed towards the end of the academic year but before the accreditation visit. The survey will also be held on an online survey platform. Due to the difficulties in getting whole-school engagement towards the end of the school year, the survey has pragmatic evaluation items whereby it can be completed on a mobile device such as a tablet or smart phone. 

### 2.7. Sample Size

There are three schools that are part of the feasibility study. Unlike a sample size calculation, a justification is made for feasibility studies [53]. In Table 4, information about the size of the school, the type and the number of participants expected to complete the comprehensive arm of the study is presented. 

According to the Department of Education and Skills school lists, School B is one of the largest secondary level in Ireland, with 1313 enrolled students. It is also a DEIS school. Approximately 10% of secondary level students attend a DEIS designated school. Moreover, there are known socioeconomic barriers towards PA [6], therefore it is necessary to carry out this feasibility in a DEIS school environment. School A is an all-girls school. Almost one in five schools in the country are all-girls’ schools. There are many reports of girls having lower levels of PA than boys and therefore it is essential to include an all-girl’s school. School C has a slightly fewer number of students than the national average of 999 students per school. Moreover, the ratio between girls and boys is slightly higher for girls (1:1.05), whereas the national average in mixed schools tends to have fewer girls than boys (1:0.87). 

### 2.8. Data Analyses

#### 2.8.1. Quantitative Data

The data from the surveys are analysed through relevant statistical methods for the follow up data in this feasibility study. Compatible data between comprehensive and basic surveys can be used to determine the test-retest reliability of the items given that a smaller subsample of the entire school. As reliability is an important psychometric property for question items, this is carried out during the first phase of data collection. 

Students take part in the comprehensive study have their measures taken two times during the academic year. The first time takes place in autumn 2018 and the second takes place six months later during the spring 2019. Accelerometer data are transferred through the ActivPal software based on 15sec epoch. The standardised cut-offs for different types of motion; sleep, standing, light, moderate and vigorous PA are then compared at an individual level from pre- and post-test time points. Similarly, the height, weight and grip strength data is compared between the time points and used to control the differences in accelerometer data. Comprehensive survey data is also analysed with differences in PA and school related factors. 

Exploratory approaches include cross-sectional multivariate analyses of PA and school-related factors as independent variables and device-based PA and perceptions of PA opportunities as the dependent variables. Mixed models and multi-level regression analyses can be used on the data that has sufficient follow up data from the first time point. The multi-level approach takes into account between- and within- individual processes that explain variances in the outcome measures. Through this approach, it is possible to test the extent of PA (psychosocial variables) and school-related factors in relation to changes in PA levels and opportunities, at the same time to examine the individual versus the school factors that contribute to the outcome variables. 

The follow-up data adds another level of analysis that can test the changes through the intervention. It makes it possible to examine, for example, the changes in PA levels across the schools from the beginning and the end of the study, while also taking into account changes in the psychosocial variables included in this study. The interactions between the contexts can confirm behavioural change theories by examining the mediating and moderation mechanisms in PA levels. The majority of the statistical analysis would be carried out using IBM SPSS. 

#### 2.8.2. Qualitative Data

The majority of the qualitative data comprises of focus group data. The way data is captured is a summary of individuals who collectively agree and discuss on the content [54]. Therefore, the first phase of analysis is to provide quantitative analysis of the subjects and the group types [55]. Focus groups can be useful to find a consensus on a phenomenon, as well as to engage with participants to discuss and share ideas that would otherwise be difficult to gather from one to one interviews [56]. In particular, the structural approach to children’s group research can be used and transferred across to adolescents so that the students’ voice to be heard [57]. Because the way a person in the focus group may consider a way to respond to the moderators’ questions could differ from what other individuals may be thinking at the time, it is important to consider the way individuals respond, with whom and in what ways [55]. Transcriptions are matched with assistant moderator notes of verbal and non-verbal behaviours. 

The data from one-to-one interviews is more straightforward. A semi-structure interview guide is used to direct the respondent to focus on the research questions and is used for further probing into these questions if the respondent needs to explain something further. Interviewees data are also merged with intonation coding to help reinforce the importance of non-verbal behaviour. The double coding from the transcription across the different qualitative approaches creates a rich source of data. 

The combination of data is inserted into NVivo software for qualitative analysis. The metadata and types of data are used to create a rich data set. The data undergoes a thematic analysis as suggested by Lederman [58] by (1) identifying the big ideas, (2) creating units of data, (3) categorizing the units, (4) negotiating categories and (5) identifying themes and use of theory. The theories surrounding social-cognitive theories, including self-efficacy theory [29], self-determination theory [31] and competence motivation theory [50] are lens used in the final steps of the content analyses. 

The data are collected through follow up measures throughout the year. The researchers incorporate verification checking at the beginning of each session to place a point where the respondents can focus on. In particular, we are interested in the processes of the intervention, as well as the potential transformation in beliefs, thoughts and actions over the course of the year. These steps are useful for designing the results in a way that allows for multi-method approach to the overall research questions. 

#### 2.8.3. Mixed Methods Analyses

Both quantitative and qualitative data can complement each other. We hope that the data that derives from both methods of inquiry can be partly explained through the literature to date and other types of data that is collected. To return to the points of evaluation of the feasibility study, there are various numbers of expected outcomes that the school is expected to achieve and they are measured directed through particular sources (Table 2). For example, the expected outcome of a broad physical education curriculum is measured through the whole school survey on participation of various physical education activities. The data taken from the beginning of the year gives insight to the types of activities that the students reported to have attended in the past 12 months. Through data collection across all year groups, the survey data can be used to determine how broad the physical education programme actually is. The post-test survey would give an indication of the extent of the physical education programme. However, reliance solely on this measure may be limited to the actual item that is included in the survey [59]. Therefore, combining the data from focus groups by the students and staff at the school can give more details about what was popular, who experienced the changes and the mechanisms in place to make the broader physical education opportunities. Therefore, the focus on the results are on the processes of creating the change, thus allowing further insight into the behavioural change techniques used to facilitate such changes. 

The SLASF log data contains both quantitative and qualitative data and can be analysed for the percent of completion towards the SLASF. Actions in relation to SLASF throughout the year form descriptive feasibility analyses. Differences in the PA audit across the year are analysed through descriptive statistics over time. In combination with the logbook of actions and the results of the PA audit more details about the feasibility of schools’ actions from the TY class can be determined in relation to desired outcomes. 

### 2.9. Availability of Data and Materials 

After completion of the study, data will be stored at the University of Limerick’s Data archive without potential identifiers and request for data can be made through the study’s principal investigator (Last author). All supplementary materials for the SLASF programme including the resource pack, template logbook and accompanying resources will be available at https://osf.io/frx6t/.

### 2.10. Ethics Approval and Consent to Participate 

The study follows the principles of the Declaration of Helsinki. The study protocol has been approved by the research ethics committee of the Faculty of Education and Health Sciences, University of Limerick (ref no. 2018/10/18_EHS). Written informed consent will be sought from participating teachers, students and students’ guardians. All participants have permission to withdraw from the study at any time and data deleted if collected. In cases of important protocol changes, requests from the ethical committee will be sought for. Trial Registration: https://osf.io/keubz/register/5771ca429ad5a1020de2872e; Registered 24th September 2018; Clinical Trial Registration: NCT03847831.

## 3. Discussion

In this year-long feasibility study of the SLASF, a mixed-method approach is used to give recommendation to stop, revise or conduct a randomised control trial. The whole-school approach requires multiple stakeholders, primarily the students in secondary level schools, the TY students, the SLASF staff and its committee, as well as the management. The theories used in this paper are based on social cognitive theories and stages of change model [29,31,50]. 

Whole-school based interventions in the promotion of PA have been increasing [11,19] although the inception of the SLASF in the secondary level schools is more complicated than primary level schools. The diversity of foci at secondary level schools brings challenges towards a uniform and national programme. This is evident to date, whereby 29% of primary schools are ASF schools, whereas less than 5% of secondary level schools have this status. Therefore, a feasibility study is needed to test the readiness prior to national roll-out. 

The results from this study would be used to help inform the development of the SLASF and report the experiences of the schools in the feasibility study. Secondary outcomes from the measures carried out in the study may lead to improved understanding of the mechanisms of the promotion of PA. Moreover, the direct mapping of the stated goals of the SLASF with measures would provide evidence. Future iterations of the SLASF may include opt in by the students to take part in the SLASF TY programme, thus providing a mixture of students who are active and inactive.

The challenges to this programme include the fidelity of the year-long programme. Schools are dynamic systems all with different characteristics based on the people who attend it. Challenging aspects could include issues arising from the coordination of the staff and pupils to carry out the tasks. There may be other activities that take place in the school, which reduce the efforts needed to run the programme or conversely, highly engagement that roles are dispersed more than previously planned. Monitoring of fidelity and carrying out process evaluations would help inform the way the programme is run.

This feasibility study is novel in design in that it a whole-school approach to the promotion of physical activity among adolescents who are empowered to organize activities over the course of the year. School management also receive an incentive by striving towards the goal and recognition of an Active School Flag. Successful piloting of the SLASF can lead to upscaling to all secondary level schools around the Republic of Ireland due to the programme endorsement by the Department of Education and Skills. Testing of the programme can be part of large scale RCT that would fit under the Healthy Ireland Demonstration Project. 

## Figures and Tables

**Figure 1 jfmk-04-00016-f001:**
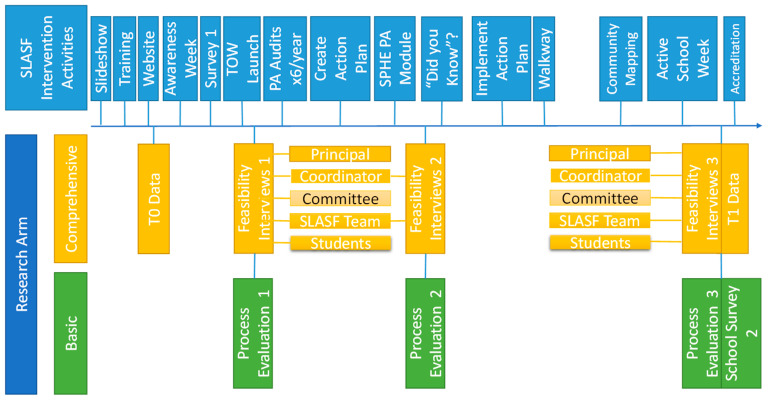
Second Level Active School Flag (SLASF) Intervention and Research Components Timeline.

**Figure 2 jfmk-04-00016-f002:**
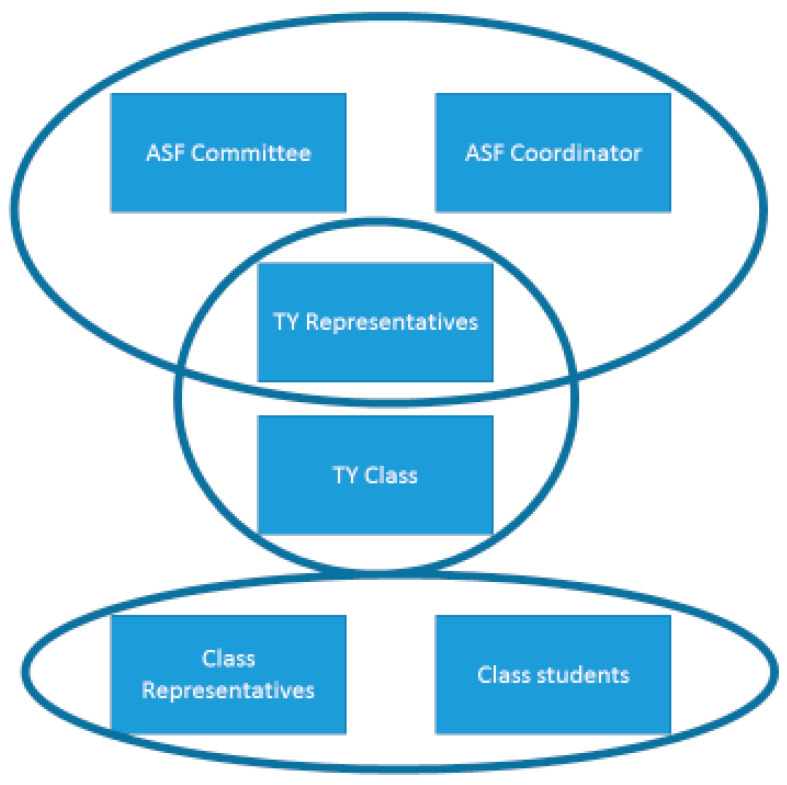
Actors of the Second Level Active school Flag.

**Table 1 jfmk-04-00016-t001:** Areas of focus for feasibility studies with measures and stopping criteria.

Focus	Measures	Stopping Criteria
Acceptability	Pragmatic survey instrument on suitability, satisfying and attractiveness for SLASF in schools for the TY and the whole-school at the end of the year.	When less than half the school students and TY consider SLASF to be suitable, satisfying or attractive to them.
	Focus group and interviews to describe the process of satisfaction of the programme, fit into the school’s culture and the positive and negative effect on the school.	When interviews describe strong statements that have a negative impact on the school’s culture or too much dissatisfaction to the programme.
Demand	PA audit of sections of the school.	No improvement over the academic year.
	Attendance list at the specific activities	Unsustainable numbers in attendance.
	Survey instrument based on readiness for PA behaviours.	Over half of students increased their readiness if they can.
	Focus groups based on the TY engagement in the SLASF process.	Discussions confirm low attendance rates and lack of demand for activities.
Implementation	Action Logs from the logbook.	Over half actions left incomplete
	Focus groups on implementation ease	Discussions where respondents report too many challenges preventing implementation
	Staff interviews	Data that suggests lack of resources to implement the programme and failures to execute the actions.
Practicality	Action plans for promotion of PA	If action plans could not be drawn up by the specified time frame
Adaptation	Registrations at events carried out as part of the action plan	Substantial decrease in participation over a 6 week period
Integration	Pre- and post-test results on PA opportunities and its participation	No increased opportunities since the beginning of the programme.
	Interviews with management about costs to organization and school policies	Descriptions whereby the costs are not sustainable. Indicators that there is a lack of staff
Expansion	Interviews of management	Descriptions of the uniqueness of the programme to the school and difficulties to roll out to other schools.
Limited Efficacy	Pre- and post-test results from the comprehensive surveys	Reduction in main outcome variables over the course of the year that is greater than the difference between each year cohort at the baseline measurements
	ASF logbook entries	Notes that report barriers to completing actions that are not manageable
	PA audit	Low level of usage when compared with the beginning of the year

**Table 2 jfmk-04-00016-t002:** Aligning outcomes with measures.

SLASF Target Areas	Measures of Efficacy	Details
Physically educated	Self-efficacy in PA	Student survey
Physically active school community	Comprehensive Survey and focus groups	Survey item and discussions
Broad physical education	Whole-school survey and focus groups	List of physical education activities and discussions
Balanced physical education	Teacher scheme of work	Teacher records
More inclusiveness	Teacher scheme of work	Teacher records
Partnership with others to promote pa culture	TY logbook	Taster session from ASW
Active school week	TY logbook	Record of entries
Increased concentration	Harter scale	Student survey
Improved learning	Teachers perceptions	Student and teacher survey
Maintenance of discipline	Teacher records	Discipline records
Improved test results	Winter and summer test	Academic records
School enjoyment	Questionnaire	Student survey
Increase Daily PA	Accelerometers	Comprehensive Data
Reduced sitting time	Accelerometers	Comprehensive Data
Reduction in overweight and obesity	Self-report and anthropometric measures of height and weight	Comprehensive data

**Table 3 jfmk-04-00016-t003:** Battery of questionnaires.

Battery	Items	Response Scales	Psychometric Information
PA Screening measure	2 items on number of days in a week of at least 60 min of MVPA per day	0–7 days	Validity & Reliability [5,44]
PA opportunities	Modified items about local opportunities for PA to the context of schools instead of ‘residential area’	5 point scale, 1 = disagree a lot, 5 = agree a lot	Original items used from an interview guide.
The exercise self-efficacy scale for adolescents	10 items on confidence to participate in a variety of conditions	11 point sliding scale, 0 = not at all confident, 10 = very confident	Nigg & Courneya, 1999 [45]
PA peer support scale	4 items on the frequency of peers influence for PA	0 = never, 5 = every day	Prochaska et al., 2002 [46]
PA, plans, expectancy and intention	Modified 3 items on the planning, expectancy and intention to do PA in the coming week	1 = unlikely, 8 = likely	Hagger et al., 2001 [47]
Readiness for behaviour change	Single item to determine which stage of the transtheoretical model in terms of PA	Select one item of each stage of the transtheoretical model	Lee et al., 2001 [48]
Perceived school performance	Single item about perceptions of teacher’s evaluation of students’ grade	Very good, good, average, below average	Felder-Puig et al., 2012 [49]
Perceived school performance	Two items about the students perception of their school grades	5 point scale, 1 = strongly disagree, 5 = strongly agree	Felder-Puig et al., 2012 [49]
Harter’s Self-perception scale for adolescents	5 items from the scholastic competence subscale.	Polarised responding	Harter et al., 1982 [50]
Belonging in school	2 items on belonging to a school	1 = Strongly agree, 5 = strongly disagree	OECD
School satisfaction	How do you feel about school a present	1 = I like it a lot, 5 = I don’t like it at all	HBSC since 2001
School effort	How pressured do you feel by the schoolwork you have to do	Not at all A little, Some, A lot	HBSC since 2001
Participation of organised activities	3 items about the student-led activities at school.	1 = Strongly agree, 5 = strongly disagree	HBSC in 2013/14
Kidscreen-27	Items on the physical and psychological well-being and the autonomy and parent relations	Not at all, Slightly, Moderately, Extremely	Ravens-Sieberer, et al., 2006 [51]

**Table 4 jfmk-04-00016-t004:** Sample size descriptions.

School	Students	Girls	Boys	Teachers	DEIS	Comprehensive (N)
Intervention						
A	971	971	-	70	N	100
B	863	440	423	70	N	150
C	1313	664	649	120	Y	150
Control						
D	582	582	-	41	N	123
E	629	322	307	45	N	121
F	378	206	172	35	Y	88

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
