# Peer review of "Feasibility Study of the Secondary Level Active School Flag Programme: Study Protocol"

_jfmk, 2019, doi:10.3390/jfmk4010016_

Round 1
Reviewer 1 Report
The manuscript illustrates the study protocol of a multi-component intervention programme at school level aimed to increase active lifestyles in Irish adolescents.
In considering that physical activity behaviours are a complex phenomenon affected by many determinants, the study protocol includes multiple data collections, which will be analysed separately and in conjunction. Overall, the protocol is based on a sound theoretical background and methodology, it is well written and of interest for the readers of the Journal. Finally, future findings have the potential to impact also at policy level.
To implement their work, I suggest the authors to consider the following:
The keywords are missing
In the introduction and/or in the discussion, the importance and modifiability of determinants of physical activity behaviours in youth should be considered. In this respect, I suggest the reference Condello et al. BMC Public Health (2016) 16:1145 DOI 10.1186/s12889-016-3800-8, which provides a comprehensive list of determinants, including the eminence on their modifiability and population level effect.
L136. I think it is most appropriate to delete “is” before “restricted”. However, I leave this aspect to the authors
Table 1. Table 1 should be implemented and formatted as Tables 2 and 3.
L179. The authors should briefly describe the “Delivering Equality of Opportunity in Schools’ (DEIS) “category.
Author Response
We would like to thank reviewer 1 to have provided a good summary and assessment of the paper. We have also worked on the areas which the reviewer pointed out, in particular reference to including keywords, a short description of DEIS, and the recommendations for the minor changes. We do recognise the importance of the paper by Condello and colleagues, but we did not include in the list of reference because there were other suitable references already in the paper.
Reviewer 2 Report
The authors just present a study protocol. There is no results. As clearly stated in the guidelines for authors, "Articles: Original research manuscripts. The journal considers all original research manuscripts provided that the work reports scientifically sound experiments and provides a substantial amount of new information", this manuscript should not be considered for publication in JMFK.
Author Response
Thank you for the review.
Reviewer 3 Report
GENERAL COMMENTS
In my opinion, the topic of this manuscript does not really fall within the scope of the Journal of Functional Morphology and Kinesiology. In an International Journal of Environmental Research and Public Health you can easily find some interesting Special Issues, like (among others) “School Health and Wellbeing”, “Physical Activity and Health” or “Health and Wellbeing of Children, Adolescents and Young Adults”. I think that Journal might be more suitable for your study.
This study examined the feasibility of the pilot programme of the Department of Education and Skills second level Active School Flag (SLASF). Three schools around Ireland were recruited as pilot schools to carry out the year-long SLASF programme with 17 planned actions involving the entire school. Data collections consist of physical measures, survey data and interviews at the beginning and the end of the academic year. The first step of the study: the feasibility of the programme through whole-school surveys, focus group discussions of key stakeholder groups and one-to-one interviews with a member of engagement and the SLASF coordinator of the school. The second step of the study: PA and its social cognitive theories based on correlates trough physical health measures, surveys carried out pre- and post-intervention, focus group discussions of the students. To sum up, in this kind of research, the study protocol is needed and highly recommended to improve the development of the SLASF programme. Thanks to the study protocol Authors described the need, relevance and priority for their study. However, despite many strengths, the article has also some weak points (corrections to text editing) which need to be improved in order to be published. These issues are presented below.
I have a feeling that your year-long programme has lots of task in a short period (17 activities are scheduled throughout the school year). Based on that, did Authors define any challenges for researchers, schools or students?
Are there any socio-cultural issues that need to be addressed in the protocol? If yes, please highlight them.
Did Authors define any balancing risks and benefits for research participants? Please ensure in the protocol that the rights of participants are protected.
Did Authors define any inclusion and exclusion criteria (age, disability, sex, place of residence and so on)?
Are you planning to conduct a follow-up study? If yes, please explain how.
SPECIFIC RECOMMENDATIONS FOR REVISION
Page 1, line 9: Please, use only lower case characters in your email address.
Page 1, line 13: Please ,use only lower case characters in your email address.
Page 1, line 21: Please, add the full name of the ASF, which is “Active School Flag”.
Page 1, line 24: What kind of school?
Page 1, line 24: Double-space between “school” and “students”. Please change it.
Page 1, line 32: Please, add the age of the students. If not here, maybe in another place of the abstract.
Page 1, lines 38-39: Please, add appropriate keywords for your study and selected Journal.
Page 1, line 42: Please, add single space between “-“ and “health”.
Page 1, line 43: Please, add reference/references here.
Page 2, line 44: Please, add single space between “-“ and “good”.
Page 2, line 45: Sometimes Authors did use “-“, another time “–“. Please, be consistent in the whole manuscript.
Page 2, line 48: Please, add single-space between “60” and “mins”.
Page 2, line 50: Double-space between “(3)” and “similarly”. Please, change it.
Page 2, line 53: Double-space between “(6)” and “global”. Please, change it.
Page 2, line 64: Please, add the abbreviation to Physical Education”, which is PE, and use it in the whole text. I have noticed that Authors are not consistent in using the abbreviation of PE.
Page 2, line 66: Add some appropriate references here.
Page 2, line 66: “Some studies” for me means more than two. Please, add some references here.
Page 2, lines 70-71: Please, add some references.
Page 2, lines 77-79: Please, add more references.
Pages 2-3, lines 94-95: Did Authors publish somewhere those results? If yes, please add references.
Page 3, line 102: Double-space between “teachers” and “in”. Please, change it.
Page 3, line 104: Double-space between “week” and “secondary”. Please, change it.
Page 3, line 107: Please, move a dot just after the reference number (21).
Page 3, line 118: Double-space between “(26)” and “many”. Please, change it.
Page 3, line 118: This is the first time in the text when I read about the GAISCE, therefore, please add the whole name to this abbreviation.
Page 3, line 119: This is the first time in the text when I read about the GAA, therefore, please add the whole name to this abbreviation.
Page 3, lines 142-143: “Studies have reported…”. Please, add more references here.
Page 4, lines 155-156: Please, add the address of the Project website to make it more clear and available.
Page 5, line 165: Please, remove one empty line in Table 1. Also, take a look at how Authors are noting the expression the “Log book” - sometimes with lower case, another time with a capital letter. In table 1, for the very first time in the text, I read about the Log book. Make sure that everything is clear and explained in the text before you will use it in the compensated version in the table. Furthermore, what mean “T0 measurements”?
Page 5, line 189: Double-space between “schools” and “this”. Please, change it.
Page 6, line 196: Double-space between “comprehensive” and “although”. Please, change it.
Page 6, line 198: Double-space between “SLASF” and “the”. Please, change it.
Page 6, line 219 and line 221: “Opt-in” or “opt in”? Please, try to be consistent.
Page 7, line 239: Double-space between “subjects” and “the”. Please, change it.
Page 7, lines 241-242: The abbreviation of DES is not clear.
Page 7, lines 246-251: I suggest to Authors to add sources (for example to provide the internet addresses).
Page 7, line 265: Why TASKS, not tasks? Please, explain.
Page 7, line 266: Please, delete the comma just after “include:”.
Page 9, line 306: Gaisce or GAISCE. Please, be consistent.
Page 10, line 339: Please, delete “Tug of War” and leave only the abbreviation, because the abbreviation has been used for the very first time on the same page (line 326).
Page 10, line 343. Please, add the dot at the end of the sentence.
Page 10, line 348: Authors mentioned that the SLASF teams are given a month to carry out analysis of the results. Are they well prepare to carry out the analysis?
Page 10, line 350 and line 353: The “principal” or Principal”. Please, be consistent.
Page 10, line 355: social and personal health education (SPHE) or Social and Personal Health Education (SPHE)?
Page 11, line 374: FUN or fun?
Page 12, line 407: 1 = disagree or 1=disagree, 1=Strongly agree or 1=strongly agree, and so one… Please double-check the column under the name “response scale” and try to be consistent there.
Page 13, line 410: A log book or a Log book? Please, be consistent.
Page 13, line 418: Please add the abbreviation (SOPLAY) just after the whole name “the System of Observing Play and Leisure Activity”.
Page 14, lines 464-465: Please, add proper references here.
Page 15, lines 509: Double-space between “(48)” and “transcriptions”. Please, change it.
Page 15, line 519: Please, change “Negotiating” to “negotiating”.
Page 16, line 548: The logbook or the Log book/ log book? Please, be consistent.
Page 16, lines 555-556: Please, add proper references here.
Page 16, lines 558-559: Any references to support this sentence?
Page 16, lines 567-568: Please, change “shy and bold”, because the meaning is unclear.
Page 16, line 566: “the opt-in” or “the opt in”. Please, try to be consistent (page 6, lines 219-221)
Page 16, lines 569-577: If possible, please include information about the novelty of your study. Authors should mention also about some practical application of this study.
Pages 17-19, lines 601-723: For bibliographic references, criteria must be unified. Please, double-check the References section and follow the Journal style.
Author Response
We would like to thank the detailed review and summary by reviewer 3. We have addressed the text editing suggestions by the reviewer. The reviewer asked if we defined any challenges for researchers, schools or students. We extended the paragraph on the challenges to the programme in the discussions section to include some possible fidelity issues. We have included a statement about the sensitivity of measures and the right for participants to not participate or their right to withdraw from the study, although the programme is designed for the entire school, hence there was no exclusion criteria from within the school. In addition, we have included some information about the planned follow-up phase in the discussion section. Finally, we have taken note and revised the manuscript according to the specific recommendations that the reviewer has diligently pointed out. We would also like to point out that we were not able to provide references about the lack of success of the ASF in the secondary level schools. However, one of the authors (KC) works for ASF and knows the figures from the central ASF database.